# The Role of the Ascorbic Acid–Glutathione Cycle in Young Wheat Ears’ Response to Spring Freezing Stress

**DOI:** 10.3390/plants12244170

**Published:** 2023-12-15

**Authors:** Yuting Zhang, Chunyang Ni, Yongwen Dong, Xue Jiang, Chang Liu, Weiling Wang, Can Zhao, Guohui Li, Ke Xu, Zhongyang Huo

**Affiliations:** Jiangsu Key Laboratory of Crop Genetics and Physiology, Jiangsu Key Laboratory of Crop Cultivation and Physiology, Jiangsu Co-Innovation Center for Modern Production Technology of Grain Crops, Agricultural College, Yangzhou University, Yangzhou 225009, China; mz120221323@stu.yzu.edu.cn (Y.Z.); mz120221369@stu.yuz.edu.cn (C.N.); mz120221361@stu.yuz.edu.cn (Y.D.); mz120231361@stu.yzu.edu.cn (X.J.); mx120200672@yzu.edu.cn (C.L.); 007148@yzu.edu.cn (C.Z.); lgh@yzu.edu.cn (G.L.); xuke@yzu.edu.cn (K.X.)

**Keywords:** wheat, young ears, spring freezing, AsA–GSH cycle, hormone

## Abstract

Freezing stress in spring often causes the death and abnormal development of young ears of wheat, leading to a significant reduction in grain production. However, the mechanisms of young wheat ears responding to freezing are largely unclear. In this study, the role of the ascorbic acid–glutathione cycle (AsA–GSH cycle) in alleviating freezing-caused oxidative damage in young wheat ears at the anther connective tissue formation phase (ACFP) was investigated. The results showed that the release rate of reactive oxygen species (ROS) and the relative electrolyte conductivity in young ears of Jimai22 (JM22, freezing-tolerant) were significantly lower than those in young ears of Xumai33 (XM33, freezing-sensitive) under freezing. The level of the GSH pool (231.8~392.3 μg/g FW) was strikingly higher than that of the AsA pool (98.86~123.4 μg/g FW) in young wheat ears at the ACFP. Freezing significantly increased the level of the AsA pool and the activities of ascorbate peroxidase (APX) and monodehydroascorbate reductase (MDHAR) in the young ears of both varieties. The level of the GSH pool increased in the young ears of XM33 under freezing but decreased in the young ears of JM22. The young ears of JM22 showed higher activities of glutathione reductase (GR), glutathione-S-transferase (GST) and glutathione peroxidase (GPX) than the young ears of XM33 under freezing. Collectively, these results suggest that the AsA–GSH cycle plays a positive role in alleviating freezing-induced oxidative damage in young wheat ears. Furthermore, the ability of utilizing GSH as a substrate to scavenge ROS is an important factor affecting the freezing tolerance of young wheat ears. In addition, abscisic acid (ABA), salicylic acid (SA), 3-indolebutyric acid (IBA) and cis-zeatin (cZ) may be involved in regulating the AsA–GSH cycle metabolism in young wheat ears under freezing.

## 1. Introduction

Wheat (*Triticum aestivum* L.) is the world’s major food crop and plays an important role in food supply security [1]. With the intensification of global warming in recent years, temperature fluctuations in spring increase the risk of wheat plants suffering from frost stress [2,3]. Spring frost is an important meteorological factor that adversely affects plant growth and development and causes serious economic losses to wheat production in China, the United States, Australia and Europe [4,5,6]. In Australia, even under optimal management conditions, spring frost can reduce wheat yields by approximately 10% [7]. The yield of wheat plants suffering from spring frost are generally reduced by 10% to 30%, and in severe cases, this can be up to 50% or more in Huanghuai and the middle and lower reaches of the Yangtze River of China [8,9]. In China, spring frost mainly occurs from March to April, when winter wheat plants are in the stage from jointing to booting [7,10]. The jointing and booting stages occur during the critical period for the rapid growth and differentiation of young wheat ears, which are very sensitive to freezing temperatures [7,11]. The death and maldevelopment of young ears are the main reasons for wheat yield reductions caused by spring freezing [12,13,14]. Zhong et al. [7] found that the freezing tolerance of young wheat ears declines as development advances through the jointing stage and drops off particularly abruptly at the point when the anther connective tissue formation phase (ACFP) begins.

Reactive oxygen species (ROS), including hydrogen peroxide (H_2_O_2_), superoxide anion (O_2_^−^), singlet oxygen (^1^O_2_) and hydroxyl radical (OH^−^), are byproducts of aerobic metabolism [15]. The inhibitory effects of freezing stress on electron transport in the membranes of mitochondria and chloroplasts and the activities of antioxidant enzymes can cause an over-accumulation of ROS in plant cells [16]. ROS are highly reactive and toxic; their over-accumulation can lead to the oxidative destruction of biological macromolecules (nucleic acids, proteins, lipids, etc.) [17]. Therefore, freezing-induced ROS over-accumulation is an important physiological cause of organ death [18,19,20]. To prevent the over-accumulation of ROS under an ever-changing environment, plants have evolved sophisticated mechanisms for scavenging ROS, including enzymatic and non-enzymatic antioxidants [21]. The ascorbate–glutathione (AsA–GSH) cycle serves as an essential antioxidant defense system in plants, which is composed of two major non-enzymatic antioxidants, AsA and GSH, together with a combination of four enzymes, ascorbate peroxidase (APX), monodehydroascorbate reductase (MDHAR), dehydroascorbate reductase (DHAR) and glutathione reductase (GR) [22,23]. Among these enzymes, APX converts H_2_O_2_ into water with the help of AsA as an electron donor, which is converted into monodehydroascorbate (MDHA). MDHAR can convert MDHA into AsA by consuming reducing power. A part of MDHA is spontaneously converted into dehydroascorbate (DHA), which can be reverted into AsA by DHAR with GSH as a substrate. GR catalyzes the reduction in GSSG for GSH regeneration [22]. In addition, glutathione peroxidase (GPX) and glutathione-S-transferase (GST) can eliminate ROS with GSH as a substrate [24]. It has been documented that the metabolism of the AsA–GSH cycle plays a crucial role in maintaining redox homeostasis in many species under drought, salt and freezing conditions [25,26]. Hormones can act as signaling molecules to regulate responses to biotic and abiotic stresses in plants [10]. Previous studies found that hormones such abscisic acid (ABA) and salicylic acid (SA) have important regulatory effects on the metabolism of the AsA–GSH cycle under stresses [20,27,28,29]. However, the responses of the AsA–GSH cycle and hormones to freezing stress in young wheat ears at the ACFP and their relationships are largely unclear.

Therefore, in this study, we analyzed the changes in the AsA–GSH cycle metabolism and hormones levels in young wheat ears’ response to freezing stress of two varieties (Jimai22, JM22, freezing-tolerant; Xumai33, XM33, freezing-sensitive) with different spring freezing tolerances at the ACFP. The results provide new insights into understanding the mechanisms of spring freezing tolerance in young wheat ears at the ACFP.

## 2. Results

### 2.1. Growth Parameters

The freezing treatment significantly inhibited the growth of wheat plants in both varieties, as indicated by the decreases in the plant height and biomass (dry mass) (Figure 1A–C). However, the decreases in the plant height and biomass in XM33 (14.89% and 27.48%) were significantly higher than that in JM22 (6.129% and 17.49%). Due to the death of the young ears from main stems or tillers caused by the freezing treatment, axillary buds sprouted to form regenerated tillers [30]. As shown in Figure 1A,D, there were more regenerated spikes (refers to those regenerated tillers that can produce grains) in XM33 than JM22 under the freezing treatment at maturity (Figure 1A,D), indicating that the freezing treatment caused more death in the young ears of XM33 than JM22. In addition, the decrease in the grain number per spike of the surviving main stems in JM22 (9.017%) was significantly lower than that in XM33 (20.63%), suggesting that there was more floret abortion in the young ears of the surviving main stems in XM33 compared to JM22 under freezing.

### 2.2. ROS Level and Electrolyte Leakage

As shown in Figure 2A, the release rate of ROS was increased in the young ears of both varieties under freezing stress compared to the normal temperature control group, with increases of 5.149% in XM33 and 2.745% in JM22. An over-accumulation of ROS can destroy the cell membrane structure and lead to electrolyte leakage. Thus, the relative electrolyte conductivity (REC) is commonly used to evaluate the freezing tolerance in plants [31]. The freezing treatment significantly increased the REC in the young ears of both varieties, indicating that the freezing treatment caused declining function in the cell membranes. However, the increase in the REC in the young ears of JM22 (45.07%) was significantly lower than that in the young ears of XM33 (100.6%) under freezing stress (Figure 2B).

### 2.3. Ascorbate and Glutathione Content

The young ears of JM22 showed significantly higher GSH (74.49%), oxidized glutthone (GSSG, 43.40%), GSH+GSSG (69.24%) and AsA (45.46%) contents, and higher GSH/GSSG and ascorbic acid/dehydroascorbic acid (AsA/DHA) ratios than those in the young ears of XM33 under normal conditions (Figure 3 and Figure 4). There were no significant differences in the DHA and AsA+DHA contents between the young ears of the two varieties (Figure 4). The freezing treatment obviously increased the contents of GSH, GSSG and GSH+GSSG, and remarkably decreased the GSH/GSSG ratio in the young ears of XM33 (Figure 3). However, the change patterns of the GSH, GSSG and GSH+GSSG contents and GSH/GSSG ratio in the young ears of JM22 under freezing stress were the opposite to those in the young ears of XM33 (Figure 3). Moreover, the freezing treatment caused an increase in the contents of AsA, DHA and AsA+DHA, as well as the AsA/DHA ratio in the young ears of both varieties (Figure 4). The young ears of JM22 exhibited significantly higher GSH (39.53%) and GSH+GSSG (22.07%) contents, as well as a higher GSH/GSSG ratio (112.9%) and a lower GSSG content (42.73%) than those in the young ears of XM33 under freezing stress (Figure 3). However, there were no significant differences in the AsA, DHA and AsA+DHA contents and the AsA/DHA ratio between the young ears of the two varieties under freezing (Figure 3 and Figure 4).

### 2.4. AsA–GSH Metabolism Associated Enzymes Activities

As shown in Table 1, the young ears of JM22 exhibited significantly higher activities of APX and GR and a significantly lower activity of GPX compared to the young ears of XM33 under normal conditions. There were no significant differences in the activities of MDHAR, DHAR and GST between the young ears of the two varieties under normal conditions. The freezing treatment enhanced the activities of MDHAR and APX and decreased the activity of GR in the young ears of both varieties. The activities of DHAR significantly increased (73.86%) in the young ears of XM33, whereas it obviously decreased (28.29%) in the young ears of JM22 under freezing stress. The change trend in the activity of GST was the opposite to that of DHAR in the young ears of the two varieties under freezing stress. The freezing treatment had no obvious effect on the GST activity in the young ears of JM22 but caused a significant decline (44.95%) in the GST activity in the young ears of XM33. The young ears of JM22 showed higher activities of GR, GST and GPX and lower activities of MDHAR, DHAR and APX compared to the young ears of XM33 under freezing stress.

### 2.5. Hormones Levels

As shown in Figure 5, the freezing treatment induced an increase in the contents of ABA (58.84% in XM33 and 50.37% in JM22), SA (82.71% in XM33 and 223.8% in JM22) and jasmonic acid (JA, 111.3% in XM33 and 28.33% in JM22) in the young ears of both varieties compared to their normal temperature control groups. The young ears of JM22 showed higher JA (87.40% and 13.79%, respectively) and SA (42.03% and 151.7%, respectively) contents and a lower ABA (21.75% and 25.92%, respectively) content than those in the young ears of XM33 under both normal and freezing conditions (Figure 5).

N6-isopentenyladenosine (IPA), trans-zeatin (tZ) and cis-zeatin (cZ) are major cytokinin derivatives in plants. The young ears of JM22 showed significantly higher contents of tZ (42.50%), cZ (146.69%) and IPA (31.04%) compared to the young ears of XM33 under normal conditions (Figure 6). The freezing treatment induced a significant decline in the contents of tZ (21.73% in XM33 and 59.90% in JM22) and IPA (25.06% in XM33 and 29.51% in JM22) in the young ears of both varieties. However, the content of cZ increased by 13.22% in the young ears of JM22 under freezing stress, while it decreased by 22.99% in the young ears of XM33. There were no significant differences in the contents of IPA and tZ between the young ears of the two varieties under freezing stress. However, the content of cZ in the young ears of JM22 was significantly higher than that in the young ears of XM33 under freezing stress (Figure 6).

There was no significant difference in the indole-3-acetic acid (IAA) content between the young ears of the two varieties under normal conditions (Figure 7). However, the young ears of JM22 showed a significantly higher 3-indolebutyric acid (IBA) content (49.60%) than that in the young ears of XM33 under normal conditions. The freezing treatment had little effect on the IAA content in the young ears of XM33, whereas it caused a significant increase in the IAA content (26.42%) in the young ears of JM22. The content of IBA was significantly decreased (48.12%) in the young ears of XM33 under freezing stress but had no significant change in the young ears of JM22. The young ears of JM22 showed higher IAA (10.76%) and IBA (209.9%) contents compared to the young ears of XM33 under freezing stress (Figure 7).

### 2.6. Correlation Analysis

As shown in Figure 8, there was a significant positive correlation between the ROS release rate and the REC in the young wheat ears under freezing, suggesting that freezing stress-induced ROS over-accumulation caused oxidative injury in the young wheat ears. The ROS release rate and REC were significantly and negatively related to the contents of AsA and GSH and the activities of GR, GST and GPX, while they were positively correlated with the activities of APX, DHAR and MDHAR in the young wheat ears under freezing. The ABA content was positively correlated with the activities of APX, MDHAR and DHA, while it was negatively related to the activities of GR, GST and GPX. In contrast, the contents of SA, IBA and cZ were positively correlated with the activities of GR, GST and GPX, while they were negatively correlated with the activities of APX, MDHAR and DHAR in the young wheat ears under freezing (Figure 8).

## 3. Discussion

The adverse effects of spring freezing on the growth and grain yield of wheat plants have been widely investigated [16,30,32,33]. Freezing stress seriously decreases the photosynthetic rates and causes a significant reduction in biomass accumulation, which can lead to declining yields [27]. Moreover, the death and abnormal development of young ears under freezing stress, resulting in reductions in the effective spike numbers and grain numbers per spike, are also important factors of spring freezing-caused yield loss [2,34,35]. The results in this study were consistent with the findings of previous studies (Figure 1). The JM22 plants showed smaller decreases in the plant height, biomass and grain numbers per spike and lower numbers of regenerated spikes compared to the XM33 plants under freezing stress (Figure 1), indicating that the JM22 plants (including the leaves and ears) were more tolerant to freezing stress than the XM33 plants. Moreover, the increase in the relative electrolyte conductivity (REC) in the young ears of JM22 was lower compared to that in the young ears of XM33 under freezing stress (Figure 2B), further demonstrating that the ears of JM22 had a better freezing tolerance than the ears of XM33 at the ACFP. Jiang et al. [3] and Wang et al. [17] reported that the content of malondialdehyde (MDA) was obviously increased in the young wheat ears at the ACFP. In the present study, the freezing treatment significantly increased the ROS release rates in the young ears of both varieties (Figure 2A). These results indicated that freezing stress caused oxidative injury to the young wheat ears, which was similar to what was observed in the wheat leaves [27,35]. The young ears of JM22 showed lower ROS release rates compared to the young ears of XM33 under freezing stress (Figure 2A), indicating that the young ears of JM22 had a stronger antioxidant capacity than the young ears of XM33. The significant positive correlation between the ROS release rate and REC (Figure 8) suggests that enhancing the antioxidant capacity is an important way to improve the freezing tolerance in young wheat ears.

Liang et al. [36] indicated that the AsA–GSH cycle was involved in maintaining redox homeostasis in wheat florets under cold stress at the booting stage. In the present study, the activity of APX in the young ears of XM33 was significantly increased under freezing stress (Table 1), suggesting that APX played a positive role in alleviating freezing-caused oxidative damage in young wheat ears at the ACFP. In addition, the activities of MDHAR and DHAR as well as the AsA/DHA ratio were significantly enhanced in the young ears of XM33 under freezing stress compared to the normal temperature control (Figure 4 and Table 1), indicating that the regeneration of AsA in the young ears of XM33 was accelerated in response to freezing stress. This result demonstrated that the regeneration of AsA plays an important role in young wheat ears resisting freezing stress. However, the generation of GSH in the young ears of XM33 was inhibited by freezing stress, as demonstrated by the decline in the GR activity and GSH/GSSG ratio (Figure 3 and Table 1). Moreover, the activities of GPX and GST in the young ears of XM33 were significantly decreased under the freezing treatment (Table 1). These results suggest that the young ears of XM33 mainly relied on AsA as a substrate to scavenge ROS rather than GSH under freezing stress at the ACFP. The increase in the GSH content in the young ears of XM33 under freezing stress might have been mainly used for the regeneration of AsA catalyzed by the enzyme DHAR [22].

The contents of AsA and DHA and the activities of MDHAR and APX were enhanced in the young ears of JM22 under freezing (Figure 4 and Table 1), implying that the young ears of JM22 also utilized AsA as a substrate to scavenge ROS under freezing. However, the young ears of JM22 displayed lower activities of MDHAR, DHAR and APX, and higher activities of GR, GST and GPX compared to the young ears of XM33 under freezing (Table 1), suggesting that the young ears of JM22 mainly depended more on GSH rather than AsA to alleviate oxidative damage caused by freezing stress compared to the young ears of XM33 at the ACFP. It is worth noting that the level of the GSH pool (231.8~392.3 μg/g FW) in the young wheat ears was strikingly higher than that of the AsA pool (98.86~123.4 μg/g FW) at the ACFP (Figure 3 and Figure 4), which was generally consistent with the results of a study on wheat florets at the booting stage [35]. These results indicated that GSH may play a more important role in maintaining redox homeostasis than AsA in young wheat ears, especially under stress conditions. In addition, the young ears of JM22 showed a significantly higher GSH content and GSH/GSSG ratio than the young ears of XM33 under freezing stress (Figure 3). It was speculated that the difference in the ability to scavenge ROS using GSH may have been an important physiological reason for the difference in the freezing tolerance between the young ears of JM22 and XM33 at the ACFP.

Although the regulatory roles of hormones in wheat leaves and rhizomes in freezing tolerance have been widely studied [37,38], their roles in the response of young wheat ears to freezing are unclear. In this study, the levels of ABA, JA and SA were significantly increased in the young wheat ears of both varieties under freezing (Figure 5), suggesting that ABA, JA and SA play important roles in regulating the freezing response of young wheat ears at the ACFP. These results were consistent with previous studies that reported that the exogenous application of ABA, SA and JA could effectively enhance the freezing tolerance of wheat plants [37,39,40]. In addition, the response patterns of cZ, IAA and IBA were different in the young ears of JM22 (increased) and XM33 (decreased) under freezing (Figure 6 and Figure 7). Zhang et al. [34] indicated that the freezing-induced abortion of florets was due to a decline in the level of IAA. The exogenous application of 6-benzylamino adenine (6-BA, a type of cytokinin) significantly increased the grain number per spike in wheat under chilling stress at the booting stage [41]. Our results indicated that cZ, IAA and IBA were positively related to the freezing tolerance of the young wheat ears at the ACFP, in accordance with the previous studies from other authors.

Kosová et al. [39] found that exogenous ABA significantly increased the AsA and GSH contents by regulating the expression levels and activities of AsA–GSH cycle metabolism-related enzymes in wheat leaves under freezing. In this study, the content of ABA was positively correlated with the activities of APX, DHAR and MDHAR, and was negatively correlated with the activities of GR, GST and GPX (Figure 8), suggesting that ABA plays opposing roles in regulating the regeneration of AsA and GSH in young wheat ears under freezing (Figure 9). These results were similar to those of Repkina et al. [42], who reported that exogenous ABA inhibited the activity of GR in pepper leaves under cold stress but increased the activity of DHAR. A previous study showed that SA, 6-BA and IAA treatments could upregulate the activities of the AsA–GSH cycle under stress conditions in plants [23,27,43]. In the present study, the contents of SA, IBA and cZ were positively correlated with the activities of GR, GST and GPX, and were negatively related to the activities of APX, DHAR and MDHAR, indicating that SA, IBA and cZ promoted the regeneration of GSH, but not AsA, in the young wheat ears under freezing stress. It is worth noting that the regulating roles of SA, IBA and cZ in the regeneration of GSH and AsA seem to be the opposite compared to ABA (Figure 8), suggesting that the metabolism of the AsA–GSH cycle is co-regulated by multiple hormones in the response of young wheat ears to freezing stress (Figure 9).

## 4. Materials and Methods

### 4.1. Experimental Design and Plant Material

This experiment was conducted during the wheat growing season of 2021–2022 at the Experimental Station of Yangzhou University, Yangzhou City, Jiangsu Province, China. Two winter wheat varieties, Xumai33 (XM33, freezing-sensitive) and Jimai22 (JM22, freezing-tolerant), were selected as the experimental materials. The wheat plants were planted in plastic pots (25 cm in diameter × 22 cm in height) filled with 9.5 kg of clay soil. The planting density was 10 plants per pot. The fertilizer and water management were carried out as described in our previous study [17]. The plants were grown outdoors until the young ears of the main stems developed to the ACFP. The development stage of the ears of the main stems in the two varieties was dynamically monitored using a microscope.

When the ears of the main stem of each variety developed to the ACFP, the wheat plants were randomly divided into two groups and moved into growth chambers (Eshengtaihe Co., Ltd., Beijing, China) for the temperature treatments. One group was transferred to a growth chamber with a temperature of 15 °C/5 °C (day/night, set according to the ambient temperature) as a control treatment (C). The other group was transferred to another growth chamber for freezing treatment (F). The freezing treatment was performed by cooling the temperature at a rate of 2 °C·h^−1^ and then holding it at −4 °C for 24 h. At the end of the freezing treatment, the ears of the main stems in each treatment were collected for further physiological and metabolic analyses. After 24 h of freezing treatment, the chamber temperature was increased to 4 °C (12 h) for thawing, and then the plants in the two growth chambers were moved out and grown under natural conditions until harvest. During the temperature treatments, the photosynthetically active radiation (PAR) in the growth chambers was set to 300 μmol m^−2^ s^−1^, the photoperiod was 10 h and the relative air humidity was maintained at approx. 70%.

### 4.2. Measurements

#### 4.2.1. Plant Height and Dry Mass

At maturity, fifteen plants were sampled and divided into three groups as three biological replicates for the measurements of the plant height. Plant height refers to the distance from the base of the stem to the top of the spike (excluding the length of the awns). Three pots of plants from each treatment were separately harvested as three biological replicates for the measurements of the aboveground biomass. The plants were killed at 105 °C for 30 min and then dried (80 °C) to a constant weight for the determination of the dry mass accumulation.

#### 4.2.2. Regenerated Tiller and Grain Number

The regenerated tillers of the three pots from each treatment were marked with small tags to distinguish them from the original tillers. The regenerated tillers that could produce grains were counted as regenerated spikes. The number of regenerated spikes and original spikes were counted upon harvest. Fifteen spikes from the main stems were collected from each treatment at maturity and were equally divided into three groups as three biological replicates for the measurements of the grain numbers per spike. Threshing and counting the number of grains per spike were carried out manually.

#### 4.2.3. ROS Release Rate and REC

The determination of the ROS release rates was accomplished using assay kits (Item No. ROS-1-Y) from Suzhou Keming Biotechnology Co., Ltd. (Suzhou, Jiangsu, China), using the method of specific probe fluorescence (2′,7′-dichlorodi-hydrofluorescin diacetate). The rate of change in the fluorescence intensity (u) was used to indicate the ROS release rate.

The REC was measured according to the method of Singh et al. [44]. Briefly, 0.1 g of fresh young ears was immersed in deionized water in test tubes for 2 h at room temperature, and the initial conductivity (E1) was measured using a conductivity bridge (DDS-307A, LEX Instruments Co., Ltd., Shanghai, China). Then, the tubes containing the young ears were boiled for 15 min to release all the electrolytes. They were cooled to room temperature, and the final conductivity (E2) was measured. The REC was calculated according to the formula REC = (E1/E2) × 100. Three replications were performed for each treatment.

#### 4.2.4. AsA–GSH Cycle

The determination of AsA, GSH, GR and GPX were accomplished using assay kits (item No. A009-1-1, item No. A006-2-1, item No. A062-1-1 and item No. A005-1, respectively) from Nanjing Jiancheng Biotechnology Co., Ltd. (Nanjing, Jiangsu, China). The measurement of DHAR, MDHAR and GSSG were accomplished using assay kits (item No. DHAR-2-W, item No. MDHAR-2-W and item No. GSSG-2-W, respectively) from Suzhou Keming Biotechnology Co., Ltd. (Suzhou, Jiangsu, China). The contents of DHA and the activity of GST were determined using kits (item No. BC1240 and item No. AKPR013U) produced by Beijing Solarbio Science and Technology Co., Ltd. (Beijing, China) and Beijing Box Bio-engineering Technology Co., Ltd. (Beijing, China), respectively. The determination of the above indicators strictly followed the manufacturer’s instructions.

The activity of APX (APX; EC1.11.1.11) was measured as described by Xia et al. [45]. In brief, 0.3 g of fresh young wheat ears were placed into a mortar, 4 mL of HEPES buffer (pH = 7.0) was added to make a homogenate, and then freeze centrifuging was applied for 20 min (10,000 r/min, 4 °C), taking 0.1 mL of the supernatant for use. To the supernatant, 2.7 mL of a reaction buffer (containing 0.1 mmol/L EDTA and 0.3 mmol/L AsA) was added sequentially, and finally 0.15 mL of H_2_O_2_ was added to initiate the enzymatic reaction, which was mixed immediately. The APX activity was calculated by recording the change in the absorbance value at 290 nm.

#### 4.2.5. Endogenous Hormone Contents

The endogenous hormone content was determined with reference to the method of Simura et al. [46]. As follows, young wheat ears were lyophilized and ground into powder. An appropriate amount of the sample was then weighed, and 1 mL of the extract (50% acetonitrile in water, precooled at −40 °C, containing isotopically labeled internal standard mixture) was added to the sample and vortexed. Then, it was mixed to make a homogenate. The homogenate was allowed to stand at −40 °C for 1 h, and then cryocentrifuged for 15 min (12,000 r/min, 4 °C). The supernatant was taken as 850 μL, which was blown dry with nitrogen and then resolubilized with 85 μL of 10% aqueous acetonitrile to obtain the endogenous hormone solution to be measured. Chromatographic separation of the various hormones was performed on a Waters ACQUITY UPLC CSH C18 (150 × 2.1 mm, 1.7 μm, Waters, Milford, MA, USA) liquid chromatography column using an ultra-high-performance liquid chromatograph (EXIONLC system, SCIEX, Framingham, MA, USA), followed by qualitative and quantitative analyses using liquid chromatography–mass spectrometry (LC–MS) in the multiple reaction monitoring (MRM) mode. In the liquid chromatography, phase A was an aqueous solution containing 0.01% formic acid, phase B was acetonitrile containing 0.01% formic acid, the temperature of the column oven was 50 °C, the sample plate was set at 4 °C and the injection volume was 5 μL.

### 4.3. Statistical Analysis

The data calculations and graphing were conducted using Microsoft Excel 2021 and Origin 2021, respectively, while the analysis of variance (ANOVA) involved SPSS 20.0. The significant differences were determined using Duncan’s multiple range test at the *p* < 0.05 level.

## 5. Conclusions

The results in this study indicated that the metabolism of the AsA–GSH cycle was involved in alleviating freezing-caused oxidative damage in young wheat ears at the ACFP. Compared to AsA, the regeneration and utilization of GSH were more important for maintaining redox homeostasis in young wheat ears under freezing stress. The hormones of SA, IBA and cZ may play a positive role in regulating the regeneration of GSH, and ABA may be implicated in controlling the regeneration of AsA in young wheat ears under freezing stress (Figure 9). However, the specific regulatory role of the various hormones in the response of young wheat ears to freezing stress and the synergistic mechanism between them need to be studied further.

## Figures and Tables

**Figure 1 plants-12-04170-f001:**
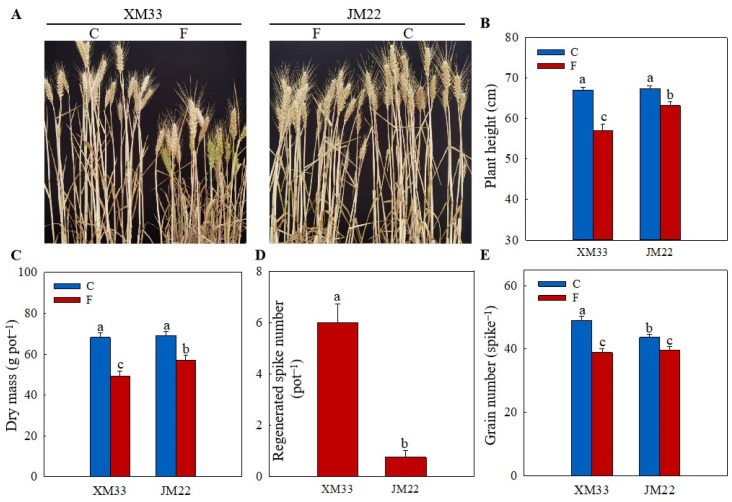
Freezing tolerances of JM22 and XM33 at the ACFP. Note: (**A**) Plant phenotype at maturity. (**B**) Plant height at maturity. (**C**) Dry mass at maturity. (**D**) Regenerated spike number. (**E**) Grain number. C and F indicate the control and freezing treatments, respectively. JM22 and XM33 represent the freezing-tolerant and freezing-sensitive winter wheat varieties Jimai22 and Xumai33, respectively. Each value is the mean ± SE of three biological replicates. The different lowercase letters indicate statistically significant differences at the *p* < 0.05 level.

**Figure 2 plants-12-04170-f002:**
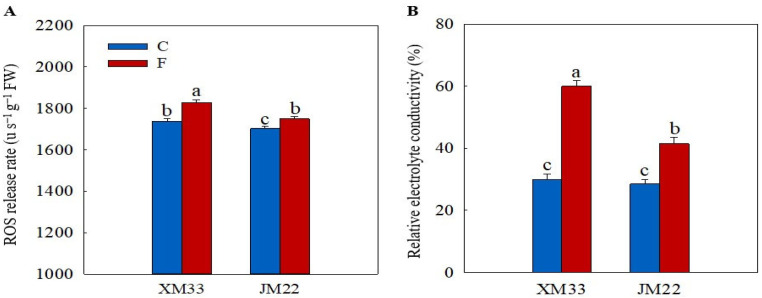
Response of the ROS release rate and relative electrolyte conductivity to freezing stress in the young wheat ears at the ACFP. Note: (**A**) ROS release rate. (**B**) Relative electrolyte conductivity (REC). C and F indicate the control and freezing treatments, respectively. JM22 and XM33 represent the freezing-tolerant and freezing-sensitive winter wheat varieties Jimai22 and Xumai33, respectively. Each value is the mean ± SE of three biological replicates. The different lowercase letters indicate statistically significant differences at the *p* < 0.05 level.

**Figure 3 plants-12-04170-f003:**
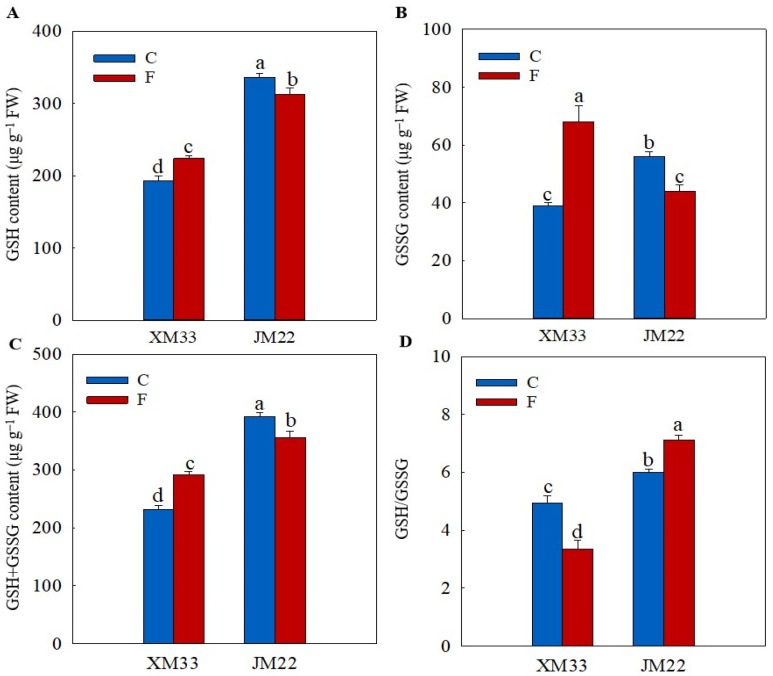
Response of the glutathione pool to freezing stress in the young wheat ears at the ACFP. Note: (**A**,**B**) Contents of reduced glutathione (GSH) and oxidized glutathione (GSSG). (**C**,**D**) Total contents of GSH and GSSG and the GSH/GSSG ratio. C and F indicate the control and freezing treatments, respectively. JM22 and XM33 represent the freezing-tolerant and freezing-sensitive winter wheat varieties Jimai22 and Xumai33, respectively. Each value is the mean ± SE of three biological replicates. The different lowercase letters indicate statistically significant differences at the *p* < 0.05 level.

**Figure 4 plants-12-04170-f004:**
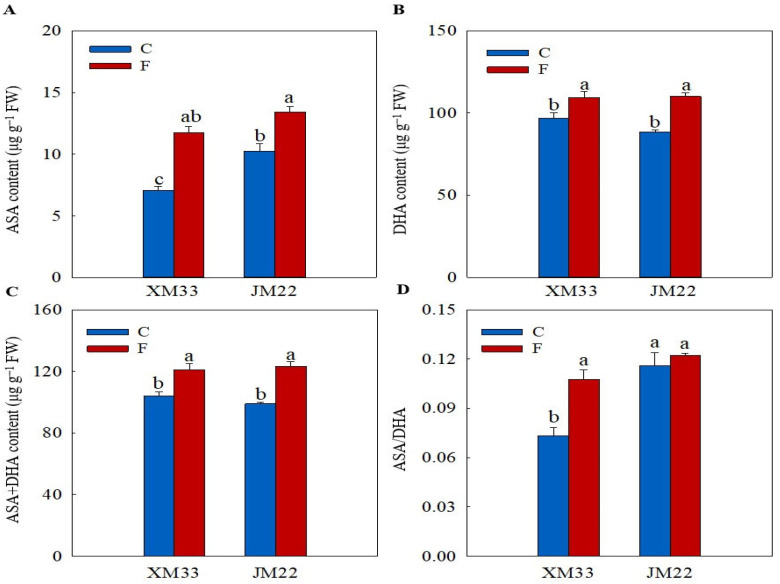
Response of the ascorbate pool to freezing stress in the young wheat ears at the ACFP. Note: (**A**,**B**) Contents of ascorbic acid (AsA) and dehydroascorbic acid (DHA). (**C**,**D**) Total contents of AsA and DHA and the AsA/DHA ratio. C and F indicate the control and freezing treatments, respectively. JM22 and XM33 represent the freezing-tolerant and freezing-sensitive winter wheat varieties Jimai22 and Xumai33, respectively. Each value is the mean ± SE of three biological replicates. The different lowercase letters indicate statistically significant differences at the *p* < 0.05 level.

**Figure 5 plants-12-04170-f005:**
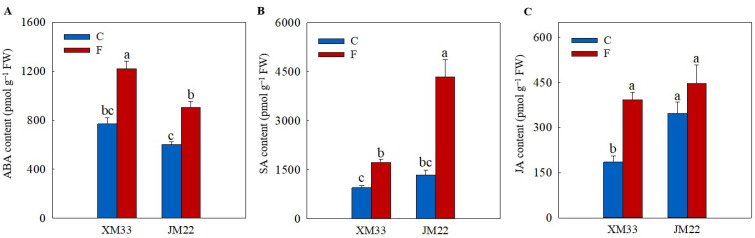
Response of the ABA, SA and JA levels to freezing stress in the young wheat ears at the ACFP. Note: (**A**–**C**) Contents of abscisic acid (ABA), salicylic acid (SA), jasmonic acid (JA). C and F indicate the control and freezing treatments, respectively. JM22 and XM33 represent the freezing-tolerant and freezing-sensitive winter wheat varieties Jimai22 and Xumai33, respectively. Each value is the mean ± SE of three biological replicates. The different lowercase letters indicate statistically significant differences at the *p* < 0.05 level.

**Figure 6 plants-12-04170-f006:**
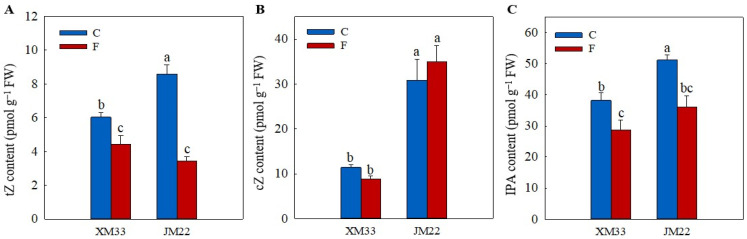
Response of the levels of cytokinin to freezing stress in the young wheat ears at the ACFP. Note: (**A**–**C**) Contents of trans-zeatin (tZ), cis-zeatin (cZ) and N6-isopentenyladenosine (IPA). C and F indicate the control and freezing treatments, respectively. JM22 and XM33 represent the freezing-tolerant and freezing-sensitive winter wheat varieties Jimai22 and Xumai33, respectively. Each value is the mean ± SE of three biological replicates. The different lowercase letters indicate statistically significant differences at the *p* < 0.05 level.

**Figure 7 plants-12-04170-f007:**
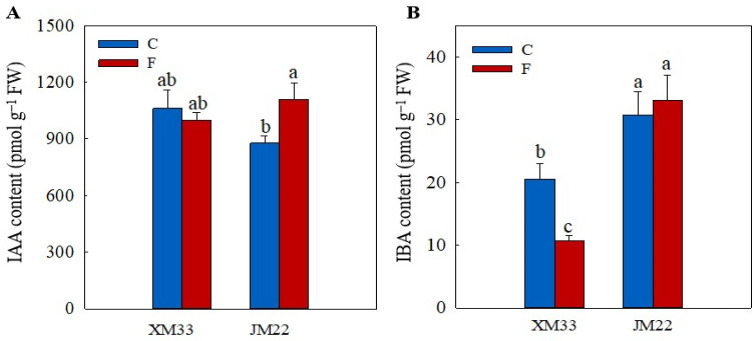
Response of the auxin levels to freezing stress in the young wheat ears at the ACFP. Note: (**A**,**B**) Contents of indoleacetic acid (IAA), indolebutyric acid (IBA). C and F indicate the control and freezing treatments, respectively. JM22 and XM33 represent the freezing-tolerant and freezing-sensitive winter wheat varieties Jimai22 and Xumai33, respectively. Each value is the mean ± SE of three biological replicates. The different lowercase letters indicate statistically significant differences at the *p* < 0.05 level.

**Figure 8 plants-12-04170-f008:**
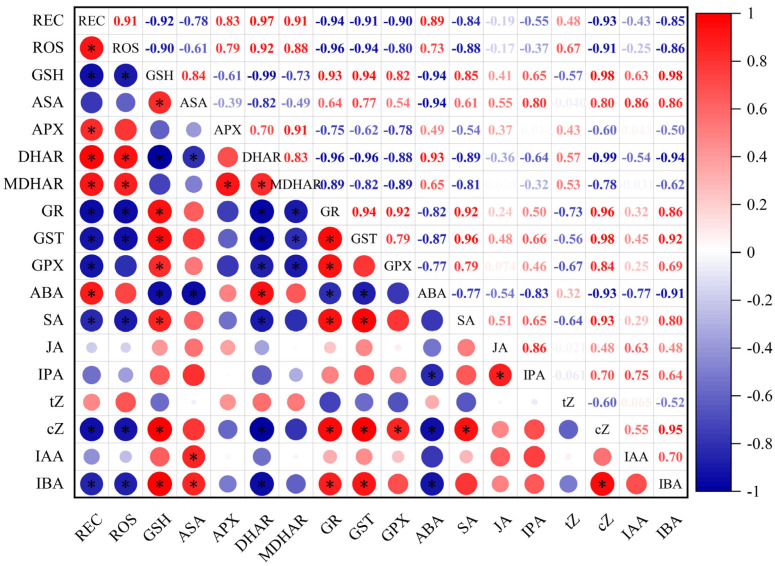
Correlation heat map for the freezing tolerance, AsA–GSH cycle metabolism and hormones levels in the young wheat ears under freezing stress at the ACFP. Note: REC, relative electrolyte conductivity; ROS, ROS release rate; GSH, the content of reduced glutathione; AsA, the content of ascorbic acid; APX, the activity of ascorbate peroxidase; DHAR, the activity of dehydroascorbate reductase; MDHAR, the activity of monodehydroascorbate reductase; GR, the activity of glutathione reductase; GST, the activity of glutathione-S-transferase; GPX, the activity of glutathione peroxidase; ABA, the content of abscisic acid; SA, the content of salicylic acid; JA, the content of jasmonic acid; IPA, the content of N6-isopentenyladenosine; tZ, the content of trans-zeatin; cZ, the content of cis-zeatin; IAA, the content of indoleacetic acid; IBA, the content of indolebutyric acid. *: Significant at the *p* < 0.05 level.

**Figure 9 plants-12-04170-f009:**
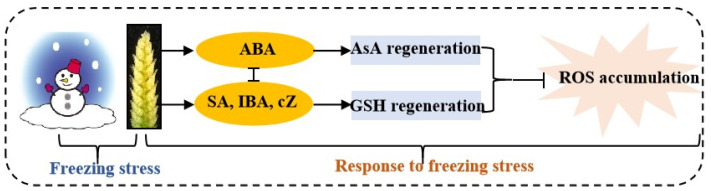
Summary showing the mechanisms of young wheat ears responding to freezing at the ACFP. Note: ABA, abscisic acid; SA, salicylic acid; IBA, indolebutyric acid; cZ, cis-zeatin; AsA, ascorbic acid; GSH, reduced glutathione; ROS, reactive oxygen species. The arrows indicate a positive regulation, whereas the lines ending with a bar show a negative regulation.

**Table 1 plants-12-04170-t001:** Response of the activities of the enzymes involved in the AsA–GSH cycle metabolism to freezing stress in the young wheat ears at the ACFP.

Enzymes	XM33		JM22	
	C	F	C	F
MDHAR (U g^−1^ FW)	2.845 ± 0.013 c	3.621 ± 0.031 a	2.822 ± 0.109 c	3.162 ± 0.145 b
DHAR (U g^−1^ FW)	0.153 ± 0.005 b	0.266 ± 0.008 a	0.152 ± 0.005 b	0.109 ± 0.005 c
APX (U g^−1^ FW)	3.233 ± 0.109 c	4.102 ± 0.023 a	3.701 ± 0.085 b	3.892 ± 0.149 ab
GR (U g^−1^ FW)	0.302 ± 0.007 b	0.189 ± 0.010 c	0.382 ± 0.010 a	0.304 ± 0.009 b
GST (U g^−1^ FW)	0.436 ± 0.014 a	0.240 ± 0.008 b	0.437 ± 0.014 a	0.417 ± 0.011 a
GPX (U g^−1^ FW)	1.211 ± 0.0323 a	1.042 ± 0.044 b	0.957 ± 0.014 b	1.204 ± 0.024 a

Note: The measurements were taken at the end of the temperature treatments. MDHAR, monodehydroascorbate reductase; DHAR, dehydroascorbate reductase; APX, ascorbate peroxidase; GR, glutathione reductase; GST, glutathione-S-transferase; GPX, glutathione peroxidase. C refers to the normal temperature control, F refers to the freezing treatment. JM22 and XM33 represent the freezing-tolerant and freezing-sensitive winter wheat varieties Jimai22 and Xumai33, respectively. The data are the means ± SE of three biological replicates. The different letters at the same row indicate significant differences at the *p* < 0.05 level.

## Data Availability

The data presented in this study are available upon request from the authors.

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
