# Peer review of "The Role of the Ascorbic Acid–Glutathione Cycle in Young Wheat Ears’ Response to Spring Freezing Stress"

_plants, 2023, doi:10.3390/plants12244170_

Round 1
Reviewer 1 Report (Previous Reviewer 1)
Comments and Suggestions for Authors
The manuscript has been significantly improved. I have no more comments.
Author Response
Dear reviewer:
We appreciate your summary of the manuscript and encouraging comment. Your assistance has helped us to make this manuscript more rigorous and has given us new insights into article corrections.
Yours Sincerely,
Weiling Wang
Reviewer 2 Report (Previous Reviewer 2)
Comments and Suggestions for Authors
The focus of this new submission has substantially changed, in comparison with previous version (rejected).
It represents a positive contribution to understand freezing impact in cellular/tissue responses, antioxidant mechanisms and hormonal involvement during anther formation of wheat ears. The subject is potentially interesting for wheat breeding programs in view of climate changes.
An attempt was made to correlate parameters (Fig. 8), which contributes to results interpretation and clearly benefits the work.
Experimental design and methodology description were clearly explained and improved.
Overall language has improved.
It is now acceptable in the present form, with minor changes (indicated below)
Line 221: avoid repeating ‘in this study’ (twice in the same sentence)
Line 224: item 2.1 should be in the plural (Growth Parameters)
Lines 1363-1365 Rephrase (suggestion: Our results indicate that….at the ACFP, in accordance with previous studies from other authors)
Lines 1726-1727 The last sentence in discussion is quite ambiguous (does not give any information, plus refers to ‘…for plants to avoid overreacting…’ which has an unclear meaning.) Rephrase this sentence for ref. 44 or remove it.
Line 1729: 4.1 should be ‘Experimental design and plant material’
Review caps letters in some subtitles (2.6, 4.2.1, 4.2.2)
Fig. 9 should be included in discussion (and/or graphical abstract)
Conclusions should be shortened
Author Response
RESPONSE TO THE COMMENTS FROM REVIEWERS
General comments
The focus of this new submission has substantially changed, in comparison with previous version (rejected).
It represents a positive contribution to understand freezing impact in cellular/tissue responses, antioxidant mechanisms and hormonal involvement during anther formation of wheat ears. The subject is potentially interesting for wheat breeding programs in view of climate changes.
An attempt was made to correlate parameters (Fig.8), which contributes to results interpretation and clearly benefits the work.
Experimental design and methodology description were clearly explained and improved.
Overall language has improved.
Response: Thank you for your recognition of this article and your valuable comments. We have revised the manuscript and made the necessary changes required based on the comments. And the changed content has been marked in yellow in the manuscript.
Specific comments are addressed as below:
Comment 1: Line 221: avoid repeating ‘in this study’ (twice in the same sentence)
Response: Thank you for pointing this out.We have revised it.
Comment 2: Line 224: item 2.1 should be in the plural (Growth Parameters)
Response: We have revised it.
Comment 3: Lines 1363-1365 Rephrase (suggestion: Our results indicate that….at the ACFP, in accordance with previous studies from other authors)
Response: Thank you for your suggestions. We have restated the statement with the following revisions:
Line321-323: Our results indicated that cZ, IAA and IBA are positively related to the freezing tolerance of young wheat ears at the ACFP, in accordance with previous studies from other authors.
Comment 4: Lines 1726-1727 The last sentence in discussion is quite ambiguous (does not give any information, plus refers to ‘…for plants to avoid overreacting…’ which has an unclear meaning.) Rephrase this sentence for ref. 44 or remove it.
Response: We have revised it.
Comment 5: Line 1729: 4.1 should be ‘Experimental design and plant material’
Response: We have revised it.
Comment 6: Review caps letters in some subtitles (2.6, 4.2.1, 4.2.2)
Response: We have revised it.
Comment 7: Fig. 9 should be included in discussion (and/or graphical abstract)
Response: We have revised it.
Comment 8: Conclusions should be shortened
Response: Thank you for your comments, we have streamlined the conclusion section as follows:
Line444-452: The results in this study indicate that the metabolism of the AsA–GSH cycle was involved in alleviating freezing-caused oxidative damage in young wheat ears at the ACFP. Compared with ASA, the regeneration and utilization of GSH were more important for maintaining redox homeostasis in young wheat ears under freezing stress. The hormones of SA, IBA and cZ may play a positive role in regulating the regeneration of GSH, and ABA may be implicated in controlling the regeneration of AsA in young wheat ears under freezing stress (Figure 9). However, the specific regulatory role of various hormones in the response of young wheat ears to freezing stress and the synergistic mechanism between them need to be studied further.

This manuscript is a resubmission of an earlier submission. The following is a list of the peer review reports and author responses from that submission.
Round 1
Reviewer 1 Report
Comments and Suggestions for Authors
The manuscript deals with wheat responses to freezing stress at the level of plant metabolites. Freezing stress is not commonly taking under experimental studies, therefore, the paper is a nice improvement of state of the art. However, the manuscript has some flaws. The details are listed below:
Abstract:
L16-21: add some % changes of examined parameters between varieties
L23-25: give a general conclusion of the study
Introduction:
I suggest to start the Introduction with a background related to the economic importance of wheat as a crop.
L47-54: add introductory sentences about the ubiquitous role of antioxidant enzymes in mitigating different types of abiotic stresses (freezing, drought, pesticides, etc.). For this purpose refer to the following reference: https://doi.org/10.1016/j.chemosphere.2022.136284
Results:
Do not use the word ‘obviously’ often.
In the Table 1 indicate which variety is freezing sensitive and tolerant
L153: rephrase
L183-209: add a full name for hormones, which are given in the text for the first time
L226-227: full names for PFK and PK
Discussion:
L244-294: this part should be rewritten because some information is repeated
Materials and Methods:
L390: freezing instead cryoprocessing
L418-426: briefly describe the methodology
L439-442: rephrase and clarify
L444: what do you mean? Clarify
L452-455: briefly describe
L456: in this part indicate mobile phase composition, column type, injection volume
L477-478: briefly describe
Comments on the Quality of English LanguageModerate editing of English language required
Reviewer 2 Report
Comments and Suggestions for Authors
The aim of this work is to gain insight into underlying mechanisms of young wheat ears tolerance in response to freezing, performing a comparative physiological analysis of two varieties Xumai33 (XM33, freezing-sensitive) and Jimai22 (JM22, freezing-tolerant) under normal and freezing conditions.
This subject is potentially interesting for breeding purposes in the context of climate changes, but the work lacks focus.
The title is somehow misleading: is the study on young ears (regenerated ?) or on strategies of two wheat cultivars to cope with freezing?
The experimental design is confuse. It is unclear under which exact conditions freezing treatment was performed. Both treatments should be under the same environmental conditions, to allow temperature decrease at the time of stress application, and this should be achieved for example using two plant growth chambers with identical conditions, which does not seem the case.
The referred physiological study, using a high number of very sensitive biochemical traits such as enzymatic acitivities and hormones or signalling molecules such as salicilic acid, is not reliable unless very rigorous treatment conditions are garanteed. All these traits are extremely dependent on every environmental changes.
Some growth parameters were also described in the study, but were barely discussed.
It would also be interesting to know about young ears fertility at harvest in both cvs.
Plant material used in each analysis is also unclearly defined. (Plant were separated in parts, but which part was used and for what ?).
The authors should have more than three biological replicates per parameter (in future experiments perhaps restrict analysis and focus in only a few, but with a higher number of plants).
M&M also needs revision to improve language (avoid mixing presnt and past tenses) and shortening unnecessary descriptions on analytical protocols.
Although the study includes a high number of analysis generating many results, such results are not linked between them, and discussion does not have a conducting line.
The english language should be accurately reviewed in all the text.
Comments on the Quality of English Language
The english language should be accurately reviewed in all the text.